# ET-Flow: Equivariant Flow-Matching for Molecular Conformer Generation

**Majdi Hassan**[*1]         **Nikhil Shenoy**[*2,4]         **Jungyoon Lee**[*1]

**Hannes Stärk**[3]         **Stephan Thaler**[4]

**Dominique Beaini**[1,4]

[1]Mila & Université de Montréal
[2]University of British-Columbia
[3]Massachusetts Institute of Technology
[4]Valence Labs

## Abstract

Predicting low-energy molecular conformations given a molecular graph is an important but challenging task in computational drug discovery. Existing state-of-the-art approaches either resort to large scale transformer-based models that diffuse over conformer fields, or use computationally expensive methods to generate initial structures and diffuse over torsion angles. In this work, we introduce **E**quivariant **T**ransformer **Flow** (ET-Flow). We showcase that a well-designed flow matching approach with equivariance and harmonic prior alleviates the need for complex internal geometry calculations and large architectures, contrary to the prevailing methods in the field. Our approach results in a straightforward and scalable method that directly operates on all-atom coordinates with minimal assumptions. With the advantages of equivariance and flow matching, ET-Flow significantly increases the precision and physical validity of the generated conformers, while being a lighter model and faster at inference. Code is available https://github.com/shenoynikhil/ETFlow.

## 1 Introduction

Generating low-energy 3D representations of molecules, called *conformers*, from the molecular graph is a fundamental task in computational chemistry as the 3D structure of a molecule is responsible for several biological, chemical and physical properties (Guimarães et al., 2012; Schütt et al., 2018, 2021; Gasteiger et al., 2020; Axelrod and Gomez-Bombarelli, 2023). Conventional approaches to molecular conformer generation consist of stochastic and systematic methods. While stochastic methods such as Molecular Dynamics (MD) accurately generate conformations, they can be slow, cost-intensive, and have low sample diversity (Shim and MacKerell Jr, 2011; Ballard et al., 2015; De Vivo et al., 2016; Hawkins, 2017; Pracht et al., 2020). Systematic (rule-based) methods (Hawkins et al., 2010; Bolton et al., 2011; Li et al., 2007; Miteva et al., 2010; Cole et al., 2018; Lagorce et al., 2009) that rely on torsional profiles and knowledge base of fragments are much faster but become less accurate

---

[*] Equal contribution.
Correspondence to: `{majdi.hassan,jungyoon.lee}@umontreal.ca` and `nikhil@valencelabs.com`

38th Conference on Neural Information Processing Systems (NeurIPS 2024).

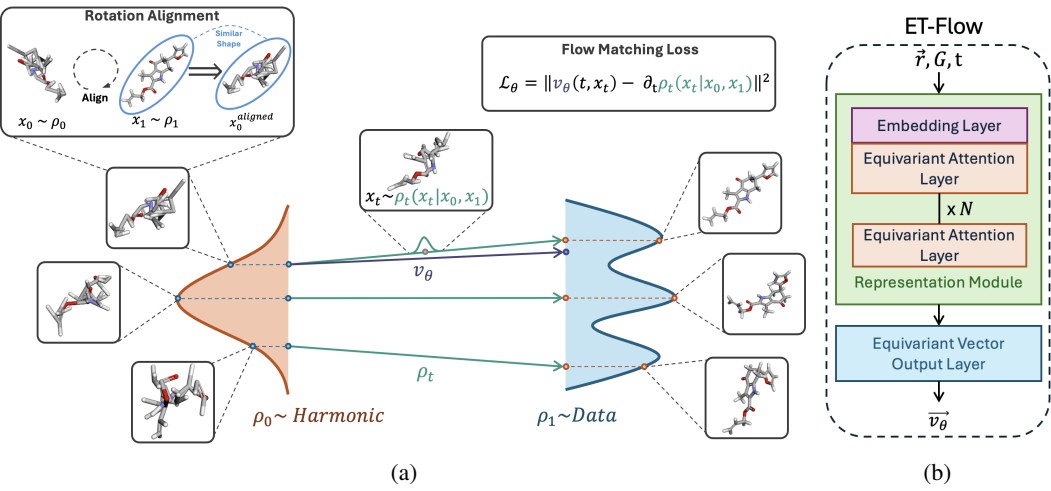

(a)                                                              (b)

Figure 1: (a) Overview of ET-Flow. The model predicts a conditional vector field $\vec{v_\theta}$ using interpolated positions ($x_t$), molecular structure ($G$), and time-step ($t$). Samples are drawn from the harmonic prior ($x_0 \sim p_0$) and then rotationally aligned with the samples from data ($x_1 \sim p_1$). A conditional probability path is constructed between pairs of $x_0$ and $x_1$, and $x_t$ is then sampled from this path at a random time $t$. (b) The ET-Flow architecture consists of a representation module based on the TorchMD-NET architecture (Thölke and De Fabritiis, 2022) and an equivariant vector output module. For detailed architecture and input preprocessing information, see Section A.1.

with larger molecules. Therefore, there has been an increasing interest in developing scalable and accurate generative modeling methods in molecular conformer generation.

Existing machine learning based approaches use diffusion models (Ho et al., 2020; Song and Ermon, 2019) to sample diverse and high quality samples given access to low-energy conformations. Prior methods typically fall into two categories: diffusing the atomic coordinates in the Cartesian space (Xu et al., 2022; Wang et al., 2024) or diffusing along the internal geometry such as pairwise distances, bond angles, and torsion angles (Ganea et al., 2021; Jing et al., 2022).

Early approaches based on diffusion (Shi et al., 2021; Luo et al., 2021; Xu et al., 2022) faced challenges such as lengthy inference and training times as well as having lower accuracy compared to cheminformatics methods. Torsional Diffusion (Jing et al., 2022) was the first to outperform cheminformatics methods by diffusing only on torsion angles after producing an initial conformer with the chemoinformatics tool RDKiT. This reliance on RDKiT structures instead of employing an end-to-end approach comes with several limitations, such as restricting the tool to applications where the local structures produced by RDKiT are of sufficient accuracy. Unlike prior approaches, the current state-of-the-art MCF (Wang et al., 2024) proposes a domain-agnostic approach by learning to diffuse over functions by scaling transformers and learning soft inductive bias from the data (Zhuang et al., 2022). Consequently, it comes with drawbacks such as high computational demands due to large number of parameters, limited sample efficiency from a lack of inductive biases like euclidean symmetries, and potential difficulties in scenarios with sparse data — a common challenge in this field.

In this paper, we propose **E**quivariant **T**ransformer **Flow** (ET-Flow), a simple yet powerful flow-matching model designed to generate low-energy 3D structures of small molecules with minimal assumptions. We utilize flow matching (Lipman et al., 2022; Albergo et al., 2023; Liu et al., 2022), which enables the learning of arbitrary probability paths beyond diffusion paths, enhancing both training and inference efficiency compared to conventional diffusion generative models. Departing from traditional equivariant architectures like EGNN (Satorras et al., 2021), we adopt an Equivariant Transformer (Thölke and De Fabritiis, 2022) to better capture geometric features. Additionally, our method integrates a Harmonic Prior (Jing et al., 2023; Stark et al., 2023), leveraging the inductive bias that atoms connected by a bond should be in close proximity. We further optimize our flow matching objective by initially conducting rotational alignment on the harmonic prior, thereby constructing shorter probability paths between source and target distributions at minimal computational cost.

Our contributions can be summarized as follows:

1. We obtain state-of-the-art precision for molecule conformer prediction, resulting in more physically realistic and reliable molecules for practitioners. We improve upon the previous methods by a large margin on ensemble property prediction.

2. We highlight the effectiveness of incorporating equivariance and more informed priors in generating physically-grounded molecules in our simple yet well-engineered method.

3. Our parameter-efficient model requires orders of magnitude fewer sampling steps than GeoDiff (Xu et al., 2022) and has significantly fewer parameters than MCF (Wang et al., 2024).

## 2   Background

**Diffusion Generative Models.** Diffusion models (Song and Ermon, 2019; Song et al., 2020; Ho et al., 2020) enables a high-quality and diverse sampling from an unknown data distribution by approximating the Stochastic Differential Equation(SDE) that maps a simple density i.e. Gaussian to the unknown data density. Concretely, it involves training a neural network to learn the score, represented as $\nabla_{\mathbf{x}} \log p_t(\mathbf{x})$ of the diffused data. During inference, the model generates sample by iteratively solving the reverse SDE. However, diffusion models have inherent drawbacks, as they (i) require on longer training times (ii) are restricted to specific probability paths and (iii) depend on the use of complicated tricks to speed up sampling (Song et al., 2020; Zhang and Chen, 2022).

**Flow Matching.** Flow Matching (Albergo et al., 2023; Lipman et al., 2022; Liu et al., 2022) provides a general framework to learn Continuous normalizing flows (CNFs) while improving upon diffusion models in simplicity, generality, and inference speed in several applications. Through simple regression against the vector field reminiscent of the score-matching objective in diffusion models, Flow matching has enabled a fast, simulation-free training of CNFs. Several subsequent studies have then expanded the scope of flow matching objective to manifolds (Chen and Lipman, 2024), arbitrary sources (Pooladian et al., 2023), and conditional flow matching with arbitrary transport maps and optimal couplings between source and target samples (Tong et al., 2023).

**Molecular Conformer Generation.** Various machine learning (ML) based approaches (Kingma and Welling, 2013; Liberti et al., 2014; Dinh et al., 2016; Simm and Hernández-Lobato, 2019; Shi et al., 2021; Luo et al., 2021; Xu et al., 2021; Ganea et al., 2021; Xu et al., 2022; Jing et al., 2022; Wang et al., 2024) have been developed to improve upon the limitations of conventional methods, among which the most advanced are TorsionDiff (Jing et al., 2022) and Molecular Conformer Fields (MCF) (Wang et al., 2024). TorsionDiff designs a diffusion model on the torsion angles while incorporating the local structure from RDKiT ETKDG (Riniker and Landrum, 2015). MCF trains a diffusion model over functions that map elements from the molecular graph to points in 3D space.

**Equivariant Architectures for Atomistic Systems.** Inductive biases play an important role in generalization and sample efficiency. In the case of 3D atomistic modelling, one example of a useful inductive bias is the euclidean group $SO(3)$ which represents rotation equivariance in 3D space. Recently, various equivariant architectures (Duval et al., 2023) have been developed that act on both Cartesian (Satorras et al., 2021; Thölke and De Fabritiis, 2022; Simeon and De Fabritiis, 2024; Du et al., 2022; Frank et al., 2022) and spherical basis (Musaelian et al., 2023; Batatia et al., 2022; Fuchs et al., 2020; Liao et al., 2023; Passaro and Zitnick, 2023; Anderson et al., 2019; Thomas et al., 2018). For molecular conformer generation, initial methods like ConfGF, DGSM utilize invariant networks as they act upon inter-atomic distances, whereas the use of equivariant GNNs have been used in GeoDiff (Xu et al., 2022) and Torsional Diffusion (Jing et al., 2022). GeoDiff utilizes EGNN (Satorras et al., 2021), a Cartesian basis equivariant architecture while Torsional Diffusion uses Tensor Field Networks (Thomas et al., 2018) to output pseudoscalars.

## 3   Method

We design ET-Flow, a scalable equivariant model that generates energy-minimized conformers given a molecular graph. In this section, we layout the framework to achieve this objective by detailing the generative process in flow matching, the rotation alignment between distributions, stochastic sampling, and finally the architecture details.

**Preliminaries** We define notation that we use throughout this paper. Inputs are continuous atom positions $\mathbf{x} \in \mathbb{R}^{N \times 3}$ where $N$ is the number of atoms. We use the notation $v_t(\mathbf{x})$ interchangeably with $v(t, \mathbf{x})$ for vector field.

## 3.1 Flow Matching

The aim is to learn a time-dependent vector field $v_t(x) : \mathbb{R}^{N \times 3} \times [0, 1] \to \mathbb{R}^{N \times 3}$ associated with the transport map $X_t : \mathbb{R}^{N \times 3} \times [0, 1] \to \mathbb{R}^{N \times 3}$ that pushes forward samples from a base distribution $\rho_0$, often an easy-to-sample distribution, to samples from a more complex target distribution $\rho_1$, the low-energy conformations of a molecule. This can be defined as an ordinary differential equation (ODE),

$$\dot{X}_t(\mathbf{x}) = v_t(X_t(\mathbf{x})), \qquad X_{t=0} = \mathbf{x}_0, \tag{1}$$

where $x_0 \sim \rho_0$. We can construct the $v_t$ via a time-differentiable interpolation between samples from $\rho_0$ and $\rho_1$ that gives rise to a probability path $\rho_t$ that we can easily sample (Lipman et al., 2022; Liu et al., 2022; Albergo and Vanden-Eijnden, 2023; Tong et al., 2023). The general interpolation between samples $x_0 \sim \rho_0$ and $x_1 \sim \rho_1$ can be defined as:

$$I_t(\mathbf{x}_0, \mathbf{x}_1) = \alpha_t \mathbf{x}_1 + \beta_t \mathbf{x}_0. \tag{2}$$

Given this interpolant that couples $\mathbf{x}_0$ and $\mathbf{x}_1$, we can define the conditional probability path as $\rho_t(\mathbf{x}|\mathbf{x}_0, \mathbf{x}_1) = \mathcal{N}(\mathbf{x}|I_t(\mathbf{x}_0, \mathbf{x}_1), \sigma_t^2 \mathbf{I})$, and the vector field can be computed as $v_t(\mathbf{x}) = \partial_t \rho_t(\mathbf{x}|\mathbf{x}_0, \mathbf{x}_1)$ which has the following form

$$v_t(\mathbf{x}) = \dot{\alpha}_t \mathbf{x}_1 + \dot{\beta}_t \mathbf{x}_0 + \dot{\sigma}_t \mathbf{z} \qquad \mathbf{z} \sim \mathcal{N}(0, \mathbf{I}). \tag{3}$$

Here we use $\dot{\alpha}_t$ as a shorthand notation for $\partial_t \alpha_t$, and similarly we apply the same notation to $\beta$ and $\sigma$. In our work, we use linear interpolation where $\alpha_t = t$, $\beta_t = 1 - t$, and $\sigma_t = \sigma\sqrt{t(1-t)}$, resulting in the vector field

$$v_t(\mathbf{x}) = \mathbf{x}_1 - \mathbf{x}_0 + \frac{1 - 2t}{2\sqrt{t(1-t)}} \mathbf{z}. \tag{4}$$

Now, we can define the objective function for learning a vector field $v_\theta(\mathbf{x})$ that generates a probability path $\rho_t$ between a base density $\rho_0$ and the target density $\rho_1$ as,

$$= \mathbb{E}_{t \sim \mathcal{U}(0,1), \mathbf{x} \sim \rho_t(\mathbf{x}_0, \mathbf{x}_1)} \|v(t, \mathbf{x}) - v_\theta(t, \mathbf{x})\|^2. \tag{5}$$

For training, we sample (i) $\mathbf{x}_0 \sim \rho_0$, $\mathbf{x}_1 \sim \rho_1$, and $t \sim \mathcal{U}(0, 1)$, (ii) interpolate according to Equation 2, (iii) add noise from a standard Gaussian, and (iv) minimize the loss defined in Equation 5. For sampling, we sample $\mathbf{x}_0 \sim \rho_0$ and integrate from $t = 0$ to $t = 1$ using the Euler's method. At each time-step, the Euler solver iteratively predicts the vector field for $\mathbf{x}_t$ and updates its position $\mathbf{x}_{t+\Delta t} = \mathbf{x}_t + v_\theta(t, \mathbf{x})\Delta t$. More details on the training and sampling algorithms are provided in Appendix B.

## 3.2 Alignment

Several previous works (Tong et al., 2023; Klein et al., 2024; Jing et al., 2024; Song et al., 2024) demonstrate that constructing a straighter path between base distribution $\rho_0$ and target distribution $\rho_1$ minimizes the transport costs and improves performance. In our work, we reduce the transport costs between samples from the harmonic prior $\rho_0$ and samples from the data distribution $\rho_1$ by rotationally aligning them using the Kabsch algorithm (Kabsch, 1976) similar to (Klein et al., 2024; Jing et al., 2024). This approach leads to faster convergence and reduces the path length between atoms by leveraging the similarity in "shape" of the samples as seen in Figure 1a without incurring high computational cost.

## 3.3 Stochastic Sampling

We employ a variant of the stochastic sampling technique inspired by (Karras et al., 2022). Specifically, we inject noise at each time step to construct an intermediate state, evaluate the vector field from the intermediate state, and then perform the deterministic ODE step from the noisy state. The original method utilizes a second-order integration, which averages the denoiser output at the noisy intermediate state and the state at the next time step after integration.

In our experiment, we use the stochastic sampler without this second-order correction term, which empirically provided a performance boost comparable to the second-order method. We apply stochastic sampling only during the final part of the integration steps, specifically within the range $t \in [0.8, 1.0]$. This helps prevent drifting towards overpopulated density regions and improves the quality of the samples (Karras et al., 2022). Stochastic sampling has improved both diversity and accuracy of the generated conformers, measured by Coverage and Average Minimum RMSD (AMR) respectively as shown in Table 1. Detailed information on the stochastic sampling algorithm is provided in algorithm B.

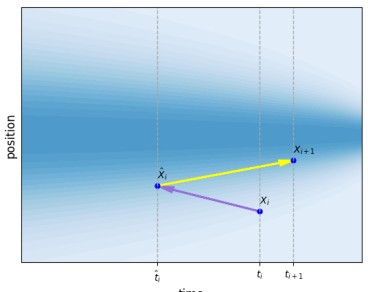

Figure 2: Stochastic sampling procedure used in inference. Noise is added to the positions $x_t$ indicated by the purple line, resulting in $\hat{x}_t$. Then, the model predicts the vector field $\hat{v}_t$ from $\hat{x}_t$ instead of $x_t$ indicted by the yellow line and updates $\hat{x}_t$ using $\hat{v}_t$ to get $x_{t+1}$.

### 3.4 Chirality Correction

While generating conformations, it is necessary to take account of the stereochemistry of atoms bonded to four distinct groups also referred to as tetrahedral chiral centers. To generate conformations with the correct chirality, we propose a simple *post hoc* trick as done in GeoMol (Ganea et al., 2021). We compare the oriented volume (OV) (Equation 6) of the generated conformation and the required orientation with the RDKit tags. In the case of a mismatch, we simply flip the conformation against the z-axis. This correction step can be efficiently performed as a batched operation since it involves a simple comparison with the required RDKit tags and an inversion of position if necessary.

$$\mathrm{OV}(\boldsymbol{p}_1, \boldsymbol{p}_2, \boldsymbol{p}_3, \boldsymbol{p}_4) = sign \left( \begin{vmatrix} 1 & 1 & 1 & 1 \\ x_1 & x_2 & x_3 & x_4 \\ y_1 & y_2 & y_3 & y_4 \\ z_1 & z_2 & z_3 & z_4 \end{vmatrix} \right). \tag{6}$$

We also consider an alternative approach for chirality correction. Instead of using the *post hoc* correction with our $O(3)$ equivariant architecture, we slightly tweak our architecture to make it $SO(3)$ equivariant by introducing a cross product term in the update layers. We compare these methods on both the GEOM-DRUGS and GEOM-QM9 dataset in Table 1 and Table 2. Our base method (ET-Flow) corresponds to using the *post hoc* correction whereas the $SO(3)$ variant is referred by ET-Flow-$SO(3)$. We empirically observe that using an additional chirality correction step is not only computationally efficient, but also performs better. We provide details on the architectural modification and proof of $SO(3)$ equivariance in Section A.1 and Section C.1 respectively.

### 3.5 Architecture

ET-Flow (Figure 1b) consists of two main components: (1) a representation module based on the equivariant transformer architecture from TorchMD-NET (Thölke and De Fabritiis, 2022) and (2) the equivariant vector output module. In the representation module, an embedding layer encodes the inputs (atomic positions, atomic numbers, atom features, bond features and the time-step) into a set of invariant features. Initial equivariant features are constructed using normalized edge vectors where the edges are constructed using a radius graph of 10 angstrom and the bonds from the 2D molecular graph. Then, a series of equivariant attention-based layers update both the invariant and equivariant features using a multi-head attention mechanism. Finally, the vector field is produced by the output layer, which updates the equivariant features using gated equivariant blocks (Schütt et al., 2018). Given that TorchMD-NET was originally designed for modeling neural network potentials, we implement several modifications to its architecture to better suit generative modeling, as detailed in Section A.1.

## 4 Experiments

We empirically evaluate ET-Flow by comparing the generated and ground-truth conformers in terms of distance-based RMSD (Section 4.2) and chemical property based metrics (Section 4.4). We

Table 1: Molecule conformer generation results on GEOM-DRUGS ($\delta = 0.75$Å). ET-Flow - SS is ET-Flow with stochastic sampling and ET-Flow - $SO(3)$ is ET-Flow using the $SO(3)$ architecture for chirality correction. For ET-Flow, ET-Flow-SS and ET-Flow-$SO(3)$, we sample conformations over 50 time-steps.

| | Recall | | | | Precision | | | |
| | Coverage ↑ | | AMR ↓ | | Coverage ↑ | | AMR ↓ | |
| | mean | median | mean | median | mean | median | mean | median |
|---|---|---|---|---|---|---|---|---|
| GeoDiff | 42.10 | 37.80 | 0.835 | 0.809 | 24.90 | 14.50 | 1.136 | 1.090 |
| GeoMol | 44.60 | 41.40 | 0.875 | 0.834 | 43.00 | 36.40 | 0.928 | 0.841 |
| Torsional Diff. | 72.70 | 80.00 | 0.582 | 0.565 | 55.20 | 56.90 | 0.778 | 0.729 |
| MCF - S (13M) | 79.4 | 87.5 | 0.512 | 0.492 | 57.4 | 57.6 | 0.761 | 0.715 |
| MCF - B (62M) | 84.0 | 91.5 | 0.427 | 0.402 | 64.0 | 66.2 | 0.667 | 0.605 |
| MCF - L (242M) | **84.7** | **92.2** | **0.390** | **0.247** | 66.8 | 71.3 | 0.618 | 0.530 |
| ET-Flow (8.3M) | 79.53 | 84.57 | 0.452 | 0.419 | 74.38 | 81.04 | 0.541 | 0.470 |
| ET-Flow - SS (8.3M) | 79.62 | 84.63 | 0.439 | 0.406 | **75.19** | **81.66** | **0.517** | **0.442** |
| ET-Flow - $SO(3)$ (9.1M) | 78.18 | 83.33 | 0.480 | 0.459 | 67.27 | 71.15 | 0.637 | 0.567 |

present the general experimental setups in Section 4.1. The implementation details are provided in Appendix A.

## 4.1 Experimental Setup

**Dataset**: We conduct our experiments on the GEOM dataset (Axelrod and Gomez-Bombarelli, 2022), which offers curated conformer ensembles produced through meta-dynamics in CREST (Pracht et al., 2024). Our primary focus is on GEOM-DRUGS, the most extensive and pharmacologically relevant subset comprising 304k drug-like molecules, each with an average of 44 atoms. We use a train/validation/test (243473/30433/1000) split as provided in (Ganea et al., 2021) Additionally, we train and test model on GEOM-QM9, a subset of smaller molecules with an average of 11 atoms. Finally, in order to assess the model's ability to generalize to larger molecules, we evaluate the model trained on GEOM-DRUGS on a GEOM-XL dataset, a subset of large molecules with more than 100 atoms. The results for GEOM-QM9 and GEOM-XL can be found in the Appendix D.

**Evaluation**: Our evaluation methodology is similar to that of (Jing et al., 2022). First, we look at RMSD based metrics like Coverage and Average Minimum RMSD (AMR) between generated and ground truth conformer ensembles. For this, we generate $2K$ conformers for a molecule with $K$ ground truth conformers. Second, we look at chemical similarity using properties like Energy ($E$), dipole moment ($\mu$), HOMO-LUMO gap ($\Delta\epsilon$) and the minimum energy ($E_{\min}$) calculated using xTB (Bannwarth et al., 2019).

**Baselines**: We benchmark ET-Flow against leading approaches outlined in Section 2. Specifically, we assess the performance of GeoMol (Ganea et al., 2021), GeoDiff (Xu et al., 2022), Torsional Diffusion (Jing et al., 2022), and MCF (Wang et al., 2024). Notably, the most recent among these, MCF, has demonstrated superior performance across evaluation metrics compared to its predecessors. It's worth mentioning that GeoDiff initially utilized a limited subset of the GEOM-DRUGS dataset; thus, for a fair comparison, we consider its re-evaluated performance as presented in (Jing et al., 2022).

## 4.2 Ensemble RMSD

As shown in Table 1 and Table 2, ET-Flow outperforms all preceding methodologies and demonstrates competitive performance with the previous state-of-the-art, MCF (Wang et al., 2024). Despite being significantly smaller with only 8.3M parameters, ET-Flow shows a substantial improvement in the quality of generated conformers, as evidenced by superior Precision metrics across all MCF models, including the largest MCF-L. When compared to MCF-S, which is closer in size, ET-Flow achieves markedly better Precision while the impact on Recall is less significant and limited to Recall Coverage. Notably, our Recall AMR remains competitive with much bigger MCF-B, underscoring the inherent advantage of our method in accurately predicting overall structures.

Table 2: Molecule conformer generation results on GEOM-QM9 ($\delta = 0.5$Å). ET-Flow - $SO(3)$ is ET-Flow using the $SO(3)$ architecture for chirality correction. For both ET-Flow and ET-Flow-$SO(3)$, we sample conformations over $50$ time-steps.

| | Recall | | | | Precision | | | |
|---|---|---|---|---|---|---|---|---|
| | Coverage ↑ | | AMR ↓ | | Coverage ↑ | | AMR ↓ | |
| | mean | median | mean | median | mean | median | mean | median |
| CGCF | 69.47 | 96.15 | 0.425 | 0.374 | 38.20 | 33.33 | 0.711 | 0.695 |
| GeoDiff | 76.50 | **100.00** | 0.297 | 0.229 | 50.00 | 33.50 | 1.524 | 0.510 |
| GeoMol | 91.50 | **100.00** | 0.225 | 0.193 | 87.60 | **100.00** | 0.270 | 0.241 |
| Torsional Diff. | 92.80 | **100.00** | 0.178 | 0.147 | 92.70 | **100.00** | 0.221 | 0.195 |
| MCF | 95.0 | **100.00** | 0.103 | 0.044 | 93.7 | **100.00** | 0.119 | 0.055 |
| ET-Flow | **96.47** | **100.00** | **0.073** | 0.047 | **94.05** | **100.00** | **0.098** | **0.039** |
| ET-Flow - $SO(3)$ | 95.98 | **100.00** | 0.076 | **0.030** | 92.10 | **100.00** | 0.110 | 0.047 |

## 4.3 Coverage Threshold Plots

We compare the coverage metrics of ET-Flow against Torsional diffusion (Jing et al., 2022) and MCF (Wang et al., 2024) against a wide range of thresholds on the GEOM DRUGS dataset in Figure 3. ET-Flow consistently outperforms previous methods in precision-based metrics. In terms of recall, our approach demonstrates better performance than Torsional Diffusion across all thresholds. Despite MCF performing better at higher thresholds, ET-Flow outperforms in the lower thresholds, underscoring its proficiency in generating accurate conformer predictions.

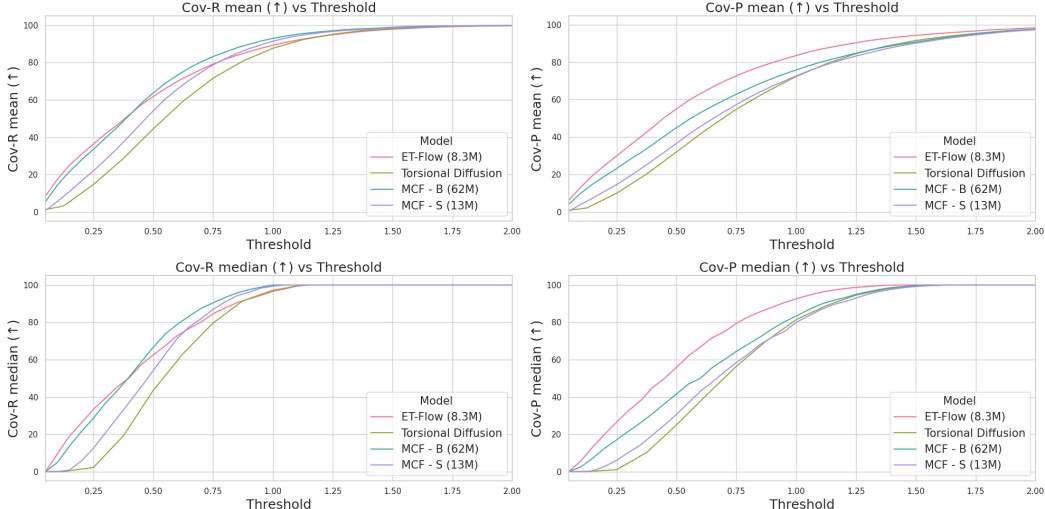

Figure 3: Recall and Precision Coverage result on GEOM-DRUGS as a function of the threshold distance. ET-Flow outperforms TorsionDiff by a large margin especially in a lower threshold region. We emphasize the better performance of ET-Flow at lower thresholds in both Recall and Precision metrics.

## 4.4 Ensemble Properties

RMSD provides a geometric measure for assessing ensemble quality, but it is also essential to consider the chemical similarity between generated and ground truth ensembles. For a random 100-molecule subset of the test set of GEOM-DRUGS, if a molecule has $K$ ground truth conformers, we generate a minimum of $2K$ and a maximum of 32 conformers per molecule. These conformers are then relaxed using GFN2-xTB (Bannwarth et al., 2019), and the Boltzmann-weighted properties of the generated and ground truth ensembles are compared. Specifically, using xTB (Bannwarth et al.,

2019), we compute properties such as energy ($E$), dipole moment ($\mu$), HOMO-LUMO gap ($\Delta\epsilon$), and the minimum energy ($E_{min}$). Table 3 illustrates the median errors for ET-Flow and the baselines, highlighting our method's capability to produce chemically accurate ensembles. Notably, we achieve significant improvements over both TorsionDiff and MCF across all evaluated properties.

Table 3: Median averaged errors of ensemble properties between sampled and generated conformers ($E$, $\Delta\varepsilon$, $E_{min}$ in kcal/mol, and $\mu$ in debye).

|  | $E$ | $\mu$ | $\Delta\epsilon$ | $E_{\min}$ |
|---|---|---|---|---|
| OMEGA | 0.68 | 0.66 | 0.68 | 0.69 |
| GeoDiff | 0.31 | 0.35 | 0.89 | 0.39 |
| GeoMol | 0.42 | 0.34 | 0.59 | 0.40 |
| Torsional Diff. | 0.22 | 0.35 | 0.54 | 0.13 |
| MCF | 0.68±0.06 | 0.28±0.05 | 0.63±0.05 | 0.04±0.00 |
| ET-Flow | **0.18±0.01** | **0.18±0.01** | **0.35±0.06** | **0.02±0.00** |

## 4.5 Inference Steps Ablation

In Table 1, our sampling process with ET-Flow utilizes 50 inference steps. To evaluate the method's performance under constrained computational resources, we conducted an ablation study by progressively reducing the number of inference steps. Specifically, we sample for 5, 10 and 20 time-steps. The results on GEOM-DRUGS are presented in Table 4. We observed minimal performance degradation with a decrease in the number of steps. Notably, ET-Flow demonstrates high efficiency, maintaining performance across all precision and recall metrics even with as few as 5 inference steps. Interestingly, ET-Flow with 5 steps still achieves superior precision metrics compared to all existing methods. This underscores ET-Flow's ability to generate high-quality conformations while operating within limited computational budgets.

## 4.6 Sampling Efficiency

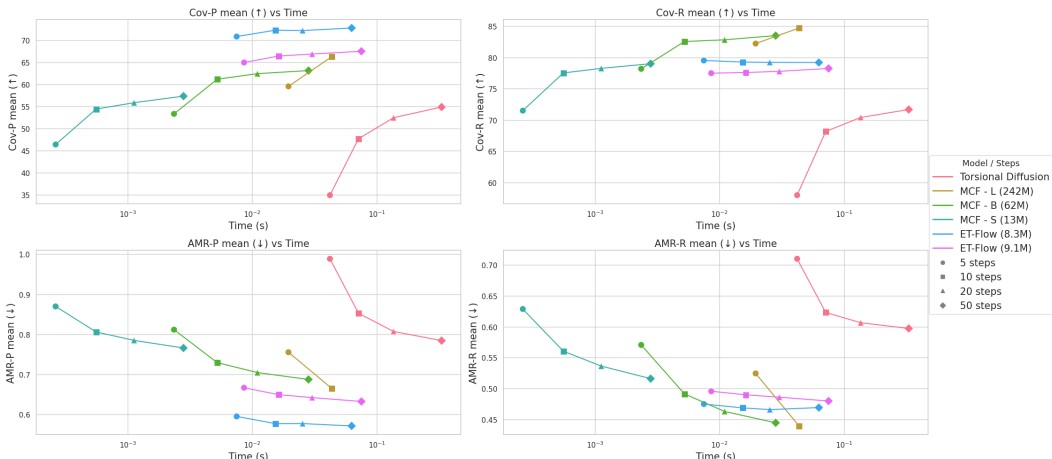

Figure 4: Sampling efficiency as a measure of the quality of Inference time with respect to the number of time steps on GEOM-DRUGS.

We demonstrate the ability of ET-Flow to generate samples efficiently. We evaluate the inference time per molecule over varying number of time steps and report the average time across 1000 random samples from the test set of GEOM-DRUGS. Figure 4 shows that ET-Flow outperforms Torsional diffusion (Jing et al., 2022) in inference across all time steps. While ET-Flow may not achieve the fastest raw inference times (potentially due to MCF variants benefiting from optimized CUDA kernels for attention), it maintains competitive speeds while ensuring higher precision. We suspect that concurrent work on improving equivariant operations with optimized CUDA kernels (Lee et al., 2024) should lead to similar efficiency gains as seen in transformer-based architectures.

Table 4: Ablation over number of inference steps on GEOM-DRUGS ($\delta = 0.75$Å). Performance of ET-Flow at 5 steps is competent across all metrics while also retaining state-of-the-art performance on precision metrics when compared with previous methods.

| | Recall | | | | Precision | | | |
|---|---|---|---|---|---|---|---|---|
| | Coverage ↑ | | AMR ↓ | | Coverage ↑ | | AMR ↓ | |
| | mean | median | mean | median | mean | median | mean | median |
| ET-Flow (5 Steps) | 77.84 | 82.21 | 0.476 | 0.443 | 74.03 | 80.8 | 0.55 | 0.474 |
| ET-Flow (10 Steps) | 79.05 | 84.00 | 0.451 | 0.415 | 74.64 | **81.38** | 0.533 | 0.457 |
| ET-Flow (20 Steps) | 79.29 | 84.04 | **0.449** | **0.413** | **74.89** | 81.32 | **0.531** | **0.454** |
| ET-Flow (50 Steps) | **79.53** | **84.57** | 0.452 | 0.419 | 74.38 | 81.04 | 0.541 | 0.470 |

ET-Flow effectively balances performance and speed, making it ideal for tasks that require high sample quality with efficient computation. With the ability to generate high-quality samples in fewer time steps, e.g., 5 time steps, as indicated in Table 4, ET-Flow is well-suited for scenarios demanding a large number of samples, as fewer steps lead to lower inference time per molecule. Additionally, we encountered difficulties running MCF-L for 20 and 50 steps, so those results have not been included. In summary, ET-Flow demonstrates efficient sampling, balancing precision and speed, making it highly effective for generating high-quality molecular samples while remaining competitive in inference time.

## 5 Conclusion

In this paper, we present our simple and scalable method ET-Flow, which utilizes an equivariant transformer with flow matching to achieve state-of-the-art performance on multiple molecular conformer generation benchmarks. By incorporating inductive biases, such as equivariance, and enhancing probability paths with a harmonic prior and RMSD alignment, we significantly improve the precision of the generated molecules, and consequently generate more physically plausible molecules. Importantly, our approach maintains parameter and speed efficiency, making it not only effective but also accessible for practical high-throughput applications.

## 6 Limitations And Future Works

While ET-Flow demonstrates competitive performance in molecular conformer generation, there are areas where it can be enhanced. One such area is the recall metrics, which capture the diversity of generated conformations. Another area is the use of an additional chirality correction step that is used to predict conformations with the desired chirality. Moreover, although our performance on the GEOM-XL dataset is comparable to MCF-S and TorsionDiff, there is still room for improvement.

We propose three future directions here. First, we observe during experiments that a well-designed sampling process incorporating stochasticity can enhance the quality and diversity of generated samples. An extension of our current approach could involve using Stochastic Differential Equations (SDEs), which utilize both vector field and score in the integration process, potentially improving the diversity of samples. Second, we propose to scale the number of parameters of ET-Flow, which has not only been shown to be useful across different domains of deep learning, but has also shown to be useful in molecular conformer generation for MCF (Wang et al., 2024). Third, to better handle the chirality problem, we aim to explore alternatives for incorporating *SO(3)*-equivariance into the model in the future.

**Acknowledgements**

The authors sincerely thank Cristian Gabellini, Jiarui Ding, and the NeurIPS reviewers for the insightful discussions and feedback. Resources used in completing this research were provided by Valence Labs. Furthermore, we acknowledge a grant for student supervision received by Mila - Quebec's AI institute - and financed by the Quebec ministry of Economy.

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

# A Implementation Details

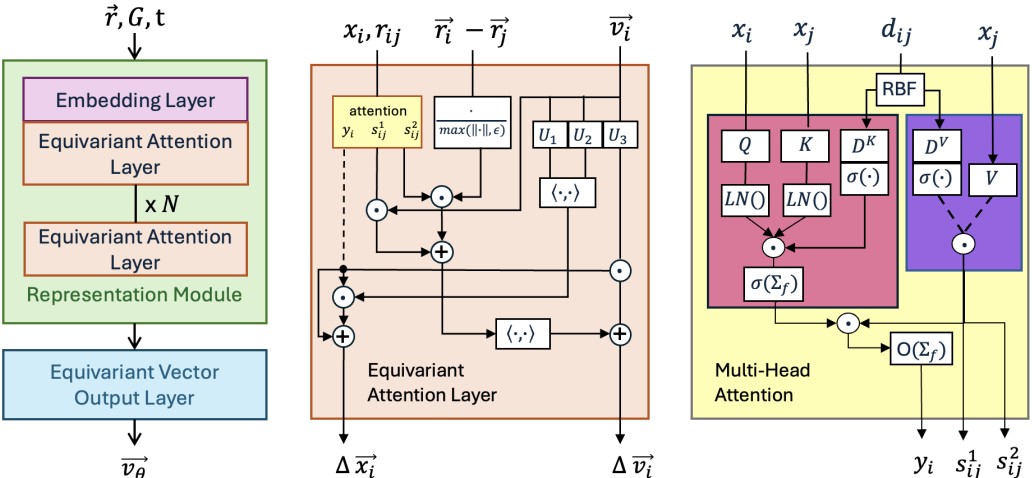

Figure 5: (a) Overall Architecture of ET-Flow consisting of 2 components, (1) Representation Layer based on TorchMD-NET Thölke and De Fabritiis (2022) and (2) Equivariant Output Layer from (Schütt et al., 2018). (b) Equivariant Attention Layer with all the operations involved, (c) Multi-Head Attention block modified with the LayerNorm.

## A.1 Architecture

The ET-Flow architecture (Figure 5) consists of 2 major components, a representation layer and an output layer. For the representation layer, we use a modified version of the embedding and equivariant attention-based update layers from the equivariant transformer architecture of TorchMD-NET (Thölke and De Fabritiis, 2022). The output layer utilizes the gated equivariant blocks from (Schütt et al., 2018). We highlight our modifications over the original TorchMD-NET architecture with this color. These modifications enable stabilized training since we use a larger network than the one proposed in the TorchMD-NET (Thölke and De Fabritiis, 2022) paper. Additionally, since our input structures are interpolations between structures sampled from a prior and actual conformations, it is important to ensure our network is numerically stable when the interpolations contain two atoms very close to each other.

**Embedding Layer**: The embedding layer maps each atom's physical and chemical properties into a learned representation space, capturing both local atomic features and geometric neighborhood information. For the $i$-th atom in a molecule with $N$ atoms, we compute an invariant embedding $x_i$ through the following process:

$$z_i = \text{embed}^{\text{int}}(z_i) \tag{7}$$

$$h_i = \text{MLP}(h_i) \tag{8}$$

where $z_i$ is the atomic number and $h_i$ represents atomic attributes (detailed in Appendix A). The MLP projects atomic attributes into a feature vector of dimension $d_h$.

Next, we compute a neighborhood embedding $n_i$ that captures local atomic environment:

$$n_i = \sum_{j=1}^{N} \text{embed}^{\text{nbh}}(z_j) \cdot g(d_{ij}, l_{ij}). \tag{9}$$

Here, $\text{embed}^{\text{nbh}}(z_j)$ provides a separate embedding for neighboring atomic numbers, $d_{ij}$ is the distance between atoms $i$ and $j$, and $l_{ij}$ encodes edge features (either from a radius-based graph or molecular bonds). The interaction function $g(d_{ij}, l_{ij})$ combines distance and edge information:

$$g(d_{ij}, l_{ij}) = W^F \left[ \phi(d_{ij}) e_1^{\text{RBF}}(d_{ij}), \ldots, \phi(d_{ij}) e_K^{\text{RBF}}(d_{ij}), l_{ij} \right] \tag{10}$$

where $e_k^{\text{RBF}}$ are $K$ exponential radial basis functions following (Unke and Meuwly, 2019), and $\phi(d_{ij})$ is a smooth cutoff function:

$$\phi(d_{ij}) = \begin{cases} \frac{1}{2}\left(\cos(\frac{\pi d_{ij}}{d_{\text{cutoff}}} + 1)\right), & \text{if } d_{ij} \leq d_{\text{cutoff}} \\ 0, & \text{otherwise} \end{cases} \tag{11}$$

Finally, we combine all features into the atom's embedding through a linear projection:

$$x_i = W^C \left[ \text{embed}^{\text{int}}(z_i), h_i, t, n_i \right] \tag{12}$$

where $t$ represents the time-step, and $[\cdot, \cdot]$ denotes concatenation. The resulting embedding $x_i \in \mathbb{R}^d$ serves as input to subsequent layers of the network.

**Attention Mechanism**: The multi-head dot-product attention operation uses atom features $x_i$, atom attributes $h_i$, time-step $t$ and inter-atomic distances $d_{ij}$ to compute attention weights. The input atom-level features $x_i$ are mixed with the atom attributes $h_i$ and the time-step $t$ using an MLP and then normalized using a LayerNorm (Ba et al., 2016). To compute the attention matrix, the inter-atomic distances $d_{ij}$ are projected into two dimensional filters $D^K$ and $D^V$ as:

$$D^K = \sigma \left( W^{D^K} e^{RBF}(d_{ij}) + b^{D^K} \right)$$
$$D^V = \sigma \left( W^{D^V} e^{RBF}(d_{ij}) + b^{D^V} \right) \tag{13}$$

The atom level features are then linearly projected along with a LayerNorm operation to derive the query $Q$ and key $K$ vectors. The value vector $V$ is computed with only the linear projection of atom-level features. Applying LayerNorm on Q, K vectors (also referred to as QK-Norm) has proven to stabilize un-normalized values in the attention matrix (Dehghani et al., 2023; Esser et al., 2024) when scaling networks to large number of parameters. The $Q$ and $K$ vectors are then used along with the distance filter $D^K$ for a dot-product operation over the feature dimension:

$$Q = \text{LayerNorm}(W^Q x_i), \quad K = \text{LayerNorm}(W^K x_i), \quad V = W^V x_i \tag{14}$$

$$\text{dot}(Q, K, D^K) = \sum_k^F Q_k \cdot K_k \cdot D_k^K. \tag{15}$$

The attention matrix is derived by passing the above dot-product operation matrix through a non-linearity and weighting it using a cosine cutoff $\phi(d_{ij})$ (similar to the embedding layer) which ensures the attention weights are non-zero only when two atoms are within a specified cutoff:

$$A = \text{SiLU}(\text{dot}(Q, K, D^K)) \cdot \phi(d_{ij}). \tag{16}$$

Using the value vector $V$ and the distance filter $D_V$, we derive 3 equally sized filters by splitting along the feature dimension,

$$s_{ij}^1, s_{ij}^2, s_{ij}^3 = \text{split}(V_j \cdot D_{ij}^V). \tag{17}$$

A linear projection is then applied to combine the attention matrix and the vectors $s_{ij}^3$ to derive an atom level feature $y_i = W^O \left( \sum_j^N A_{ij} \cdot s_{ij}^3 \right)$. The output of the attention operation are $y_i$ (an atom level feature) and two scalar filters $s_{ij}^1$ and $s_{ij}^2$ (edge-level features).

**Update Layer**: The update layer computes interactions between atoms in the attention block and uses the outputs to update the scalar feature $x_i$ and the vector feature $\vec{v}_i$. First, the scalar feature output $y_i$ from the attention mechanism is split into three features $(q_i^1, q_i^2, q_i^3)$, out of which $q_i^1$ and $q_i^2$ are used for the scalar feature update as,

$$\Delta x_i = q_i^1 + q_i^2 \cdot \langle U_1 \vec{v}_i \cdot U_2 \vec{v}_i \rangle, \tag{18}$$

where $\langle U_1 \vec{v}_i \cdot U_2 \vec{v}_i \rangle$ is the inner product between linear projections of vector features $\vec{v}_i$ with matrices $U_1, U_2$.

The edge vector update consists of two components. First, we compute a vector $\vec{w}i$, which for each atom is computed as a weighted sum of vector features and a clamped-norm of the edge vectors over all neighbors:

$$\vec{w}_i = \sum_j^N s_{ij}^1 \cdot \vec{v}_j + s_{ij}^2 \cdot \frac{\vec{r}_i - \vec{r}_j}{\max(\|\vec{r}_i - \vec{r}_j\|, \epsilon)}, \tag{19}$$

$$\Delta \vec{v}_i = \vec{w}_i + q_i^3 \cdot U_3 \vec{v}_i \tag{20}$$

where $U_1$ and $U_3$ are projection matrices over the feature dimension of the vector feature $\vec{v}_i$. In this layer, we clamp the minimum value of the norm (to $\epsilon = 0.01$) to prevent numerically large values in cases where positions of two atoms are sampled too close from the prior.

$SO(3)$ **Update Layer**: We also design an $SO(3)$ equivariant architecture by adding an additional cross product term in Equation 19 as follows,

$$\vec{w}_i = \sum_j^N s_{ij}^1 \cdot \vec{v}_j + s_{ij}^2 \cdot \frac{\vec{r}_i - \vec{r}_j}{\max(\|\vec{r}_i - \vec{r}_j\|, \epsilon)} + s_{ij}^4 \cdot \left( \vec{v}_j \times \frac{\vec{r}_i - \vec{r}_j}{\max(\|\vec{r}_i - \vec{r}_j\|, \epsilon)} \right), \tag{21}$$

where $sij^4$ is derived by modifying the split operation Equation 17 in the attention layer where the value vector $V$ and distance filter $D_V$ is projected into 4 equally sized filters instead of 3.

**Output Layer**: The output layer consists of Gated Equivariant Blocks from (Schütt et al., 2018). Given atom scalar $x_i$ and vector features $\vec{v}_i$, the updates in each block is defined as,

$$x_{i,\text{updated}}, \vec{w}_i = \text{split}(\text{MLP}([x_i, U_1\vec{v}_i])) \tag{22}$$

$$\vec{v}_{i,\text{updated}} = (U_2\vec{v}_i) \cdot \vec{w}_i \tag{23}$$

Here, $U_1$ and $U_2$ are linear projection matrices that act along feature dimension. Our modification is to use LayerNorm in the MLP to improve training stability.

## A.2 Input Featurization

Atomic features (or Node Features) are computed using RDKit (Landrum et al., 2013) features as described in Table 5. For computing edge features and edge index, we use a combination of global (radius based edges) and local (molecular graph edges) similar to (Jing et al., 2022).

| Name | Description | Range |
|------|-------------|-------|
| `chirality` | Chirality Tag | {unspecified, tetrahedral CW & CCW, other} |
| `degree` | Number of bonded neighbors | $\{x : 0 \leq x \leq 10, x \in \mathbb{Z}\}$ |
| `charge` | Formal charge of atom | $\{x : -5 \leq x \leq 5, x \in \mathbb{Z}\}$ |
| `num_H` | Total Number of Hydrogens | $\{x : 0 \leq x \leq 8, x \in \mathbb{Z}\}$ |
| `number_radical_e` | Number of Radical Electrons | $\{x : 0 \leq x \leq 4, x \in \mathbb{Z}\}$ |
| `hybrization` | Hybrization type | {sp, sp$^2$, sp$^3$, sp$^3$d, sp$^3$d$^2$, other} |
| `aromatic` | Whether on a aromatic ring | {True, False} |
| `in_ring` | Whether in a ring | {True, False} |

Table 5: Atomic features included in ET-Flow.

## A.3 Evaluation Metrics

Following the approaches of (Ganea et al., 2021; Xu et al., 2022; Jing et al., 2022), we utilize Average Minimum RMSD (AMR) and Coverage (COV) to assess the performance of molecular conformer generation. Here, $C_g$ denotes the set of generated conformations, and $C_r$ denotes the set of reference conformations. For both AMR and COV, we calculate and report Recall (R) and Precision (P). Recall measures the extent to which the generated conformers capture the ground-truth conformers, while Precision indicates the proportion of generated conformers that are accurate. The specific formulations for these metrics are detailed in the following equations:

$$\text{AMR-R}(C_g, C_r) = \frac{1}{|C_r|} \sum_{\mathbf{R} \in C_r} \min_{\hat{\mathbf{R}} \in C_g} \text{RMSD}(\mathbf{R}, \hat{\mathbf{R}})$$

$$\text{COV-R}(C_g, C_r) = \frac{1}{|C_r|} |\{\mathbf{R} \in C_r | \text{RMSD}(\mathbf{R}, \hat{\mathbf{R}}) < \delta, \hat{\mathbf{R}} \in C_g\}|$$

$$\text{AMR-P}(C_r, C_g) = \frac{1}{|C_g|} \sum_{\hat{\mathbf{R}} \in C_g} \min_{\mathbf{R} \in C_r} \text{RMSD}(\hat{\mathbf{R}}, \mathbf{R})$$

$$\text{COV-P}(C_r, C_g) = \frac{1}{|C_g|} |\{\hat{\mathbf{R}} \in C_g | \text{RMSD}(\hat{\mathbf{R}}, \mathbf{R}) < \delta, \mathbf{R} \in C_r\}|$$

A lower AMR score signifies improved accuracy, while a higher COV score reflects greater diversity in the generative model. Following (Jing et al., 2022), the threshold $\delta$ is set to 0.5Å for GEOM-QM9 and 0.75Å for GEOM-DRUGS.

## A.4 Training Details and Hyperparameters

For GEOM-DRUGS, we train ET-Flow for a fixed 250 epochs with a batch size of 64 and 5000 training batches per epoch per GPU on 8 A100 GPUs. For the learning rate, we use the Adam Optimizer with a cosine annealing learning rate which goes from a maximum of $10^{-3}$ to a minimum $10^{-7}$ over 250 epochs with a weight decay of $10^{-10}$. For GEOM-QM9, we train ET-Flow for 200 epochs with a batch size of 128, and use all of the training dataset per epoch on 4 A100 GPUs. We use the cosine annealing learning rate schedule with maximum of $8 \cdot 10^{-4}$ to minimum of $10^{-7}$ over 100 epochs, post which the maximum is reduced by a factor of 0.05. We select checkpoints based on the lowest validation error.

| Hyper-parameter | ET-Flow |
|---|---|
| `num_layers` | 20 |
| `hidden_channels` | 160 |
| `num_heads` | 8 |
| `neighbor_embedding` | True |
| `cutoff_lower` | 0.0 |
| `cutoff_higher` | 10.0 |
| `node_attr_dim` | 8 |
| `edge_attr_dim` | 1 |
| `reduce_op` | True |
| `activation` | SiLU |
| `attn_activation` | SiLU |
| `# param` | 8.3M |

Table 6: Hyperparameters for ET-Flow

# B  Training and Sampling Algorithm

The following algorithms go over the pseudo-code for the training and sampling procedure. For each molecule, we use up to 30 conformations with the highest boltzmann weights as provided by CREST (Pracht et al., 2024) similar to that of (Jing et al., 2022)

---

**Algorithm 1:** Training procedure

---

**Input:** molecules $[G_0, ..., G_N]$ each with true conformers $[C_{G,1}, ...C_{G,K_G}]$, the harmonic prior $\rho_0$, learning rate $\alpha$, number of epochs $N_e$, initialized vector field $v_\theta$

**Output:** trained flow matching model $v_\theta$

for $i \leftarrow 1$ to $N_e$ do
  for $G$ *in* $[G_0, ..., G_N]$ do
    Sample $t \sim \mathcal{U}[0, 1]$ and $C_1 \in [C_{G,1}, ...C_{G,K_G}]$;
    Sample prior $C_0 \sim \rho_0(G)$;
    Align $C_0 \leftarrow \text{RMSDAlign}(C_0, C_1)$;
    Sample $C_t = tC_1 + (1-t)C_0 + \sigma^2 t(1-t)z, \quad z \sim \mathcal{N}(0, \mathbf{I})$;
    Construct vector field $u_t \leftarrow x_1 - x_0 + \frac{1-2t}{2\sqrt{t(1-t)}}z$;
    Compute loss $\leftarrow \|v_\theta(t, C_t) - u_t\|^2$;
    Take gradient step $\theta \leftarrow \theta - \alpha\nabla_\theta$;

---

**Algorithm 2:** Inference procedure

---

**Input:** molecular graph $G$, number conformers $K$, number of sampling steps $N$

**Output:** predicted conformers $[C_1, ...C_K]$

for $C$ *in* $[C_1, ...C_K]$ do
  sample prior $\hat{C} \sim \rho_0(G)$;
  for $n \leftarrow 0$ to $N - 1$ do
    Set $t \leftarrow \frac{n}{N}$;
    Set $\Delta t \leftarrow \frac{1}{N}$;
    Predict $\hat{v} = v_\theta(t, \hat{C})$;
    Update $\hat{C} = \hat{C} + \hat{v}\Delta t$

**Algorithm 3:** Stochastic Sampler

---

**Input:** molecular graph $G$, number conformers $K$, number of sampling steps $N$, stochasticity
     level $churn$, stochastic sampling range $[t_{min}, t_{max}]$
**Output:** predicted conformers $[C_1, ... C_K]$
**for** $C$ **in** $[C_1, ... C_K]$ **do**
    sample prior $\hat{C} \sim \rho_0(G)$;
    **for** $n \leftarrow 0$ **to** $N - 1$ **do**
        Set $t \leftarrow \frac{n}{N}$;
        Set $\Delta t \leftarrow \frac{1}{N}$;
        Set $\gamma \leftarrow \frac{churn}{N}$;
        **if** $t \in [t_{min}, t_{max}]$ **then**
            Sample $\epsilon \sim N(0, I)$;
            $\Delta \hat{t} \leftarrow \gamma(1 - t)$;
            $\hat{t} \leftarrow max(t - \Delta \hat{t}, 0)$;
            $\hat{C} \leftarrow \hat{C} + \Delta \hat{t} \sqrt{t^2 - \hat{t}^2} \epsilon$;
            Predict $\hat{v} = v_\theta(\hat{t}, \hat{C})$;
            Set $\Delta t \leftarrow \Delta t + \Delta \hat{t}$;
        **else**
            Predict $\hat{v} = v_\theta(t, \hat{C})$;
        Update $\hat{C} = \hat{C} + \hat{v} \Delta t$

---

## C   Proofs

### C.1   Designing SO(3) Equivariance

We show that we can modify the architecture in Section A.1 (Equation 18) to produce a final vector output that satisfies rotation equivariance and reflection asymmetry. Let $\vec{v}_1$ and $\vec{v}_2$ be linearly independent non-zero vectors $\|\vec{v}_1\| > 0, \|\vec{v}_2\| > 0$, and $s$ be a scalar. We implement SO(3) equivariance by adding a vector with a cross product. We show that vector $\vec{v} = \vec{v}_1 + s(\vec{v}_1 \times \vec{v}_2)$, where $\vec{v}_1 \times \vec{v}_2$ denotes cross product of $\vec{v}_1$ and $\vec{v}_2$, satisfies anti-symmetry while maintaining rotation equivariance as follows,

$$R\vec{v}_1 + s(R\vec{v}_1 \times R\vec{v}_2) = R(\vec{v}_1) + sR(\vec{v}_1 \times \vec{v}_2) \tag{24}$$
$$= R(\vec{v}_1 + s(\vec{v}_1 \times \vec{v}_2)) \tag{25}$$
$$-\vec{v}_1 + s(-\vec{v}_1 \times -\vec{v}_2) = -\vec{v}_1 + s(\vec{v}_1 \times \vec{v}_2) \tag{26}$$
$$\neq -(\vec{v}_1 + s(\vec{v}_1 \times \vec{v}_2)) \tag{27}$$

This concludes the proof for rotation equivariance and reflection anti-symmetry.

## D   Additional Results

### D.1   Design Choice Ablations

We conduct a series of ablation studies to assess the influence of each component in the ET-Flow. Particularly, we re-run the experiments with (1) $O(3)$ equivariance without chirality correction, (2) Absence of Alignment, (3) Gaussian Prior as a base distribution. We demonstrate that improving probability paths and utilizing an expressive equivariant architecture with correct symmetries are key components for ET-Flow to achieve state of the art performance. The ablations were ran with reduced settings (50 epochs; $lr = 1e - 4$; 4 A100 gpus). Results are shown in Table D.1.

### D.2   Results on GEOM-XL

We now assess how well a model trained on GEOM-DRUGS generalises to unseen molecules with large numbers of atoms, using the GEOM-XL dataset containing a total of 102 molecules. This provides insights into the model's capacity to tackle larger molecules and out-of-distribution tasks. Upon executing the checkpoint

Table 7: Ablation results on GEOM-DRUGS.

| | Recall | | | | Precision | | | |
| --- | --- | --- | --- | --- | --- | --- | --- | --- |
| | Coverage ↑ | | AMR ↓ | | Coverage ↑ | | AMR ↓ | |
| | mean | median | mean | median | mean | median | mean | median |
| ET-Flow | 75.37 | 82.35 | 0.557 | 0.529 | 58.90 | 60.87 | 0.742 | 0.690 |
| ET-Flow ($O(3)$) | 72.74 | 79.21 | 0.576 | 0.556 | 54.84 | 54.11 | 0.794 | 0.739 |
| ET-Flow (w/o Alignment) | 68.67 | 74.71 | 0.622 | 0.611 | 47.09 | 44.25 | 0.870 | 0.832 |
| ET-Flow (Gaussian Prior) | 66.53 | 73.01 | 0.640 | 0.625 | 44.41 | 40.88 | 0.903 | 0.864 |

Table 8: Generalization results on GEOM-XL.

| | AMR-P ↓ | | AMR-R ↓ | | # mols |
| --- | --- | --- | --- | --- | --- |
| | mean | median | mean | median | |
| GeoDiff | 2.92 | 2.62 | 3.35 | 3.15 | - |
| GeoMol | 2.47 | 2.39 | 3.30 | 3.14 | - |
| Tor. Diff. | 2.05 | 1.86 | **2.94** | 2.78 | - |
| MCF - S | 2.22 | 1.97 | 3.17 | 2.81 | 102 |
| MCF - B | 2.01 | 1.70 | 3.03 | 2.64 | 102 |
| MCF - L | **1.97** | **1.60** | **2.94** | **2.43** | 102 |
| ET-Flow (ours) | 2.31 | 1.93 | 3.31 | 2.84 | 102 |
| Tor. Diff. | 1.93 | 1.86 | 2.84 | 2.71 | 77 |
| MCF - S | 2.02 | 1.87 | 2.9 | 2.69 | 77 |
| MCF - B | 1.71 | 1.61 | 2.69 | 2.44 | 77 |
| MCF - L | **1.64** | **1.51** | **2.57** | **2.26** | 77 |
| ET-Flow (ours) | 2.00 | 1.80 | 2.96 | 2.63 | 75 |

provided by Torsional Diffusion, we encountered 27 failed cases for generation likely due to RDKit failures, similar to the observations in MCF albeit with slightly different exact numbers. In both experiments involving all 102 molecules and a subset of 75 molecules, ET-Flow achieves performance comparable to Torsional Diffusion and MCF-S, but falls short of matching the performance of MCF-B and MCF-L. It's worth noting that MCF-B and MCF-L are significantly larger models, potentially affording them an advantage in generalization tasks. As part of our future work, we plan to scale up our model and conduct further tests to explore its performance in this regard.

### D.3 Additional Out-of-Distribution Results

Table 9: Additional OOD results. We use RS and SS to indicate Random Split and Scaffold Split respectively.

| | Recall | | | | Precision | | | |
| --- | --- | --- | --- | --- | --- | --- | --- | --- |
| | Coverage ↑ | | AMR ↓ | | Coverage ↑ | | AMR ↓ | |
| | mean | median | mean | median | mean | median | mean | median |
| ET-Flow (QM9 RS) | 96.47 | 100.00 | 0.073 | 0.047 | 94.05 | 100.00 | 0.098 | 0.039 |
| ET-Flow (QM9 SS) | 95.00 | 100.00 | 0.083 | 0.029 | 90.25 | 100.00 | 0.124 | 0.053 |
| ET-Flow (DRUGS → QM9) | 86.68 | 100.00 | 0.218 | 0.160 | 68.69 | 75.30 | 0.369 | 0.317 |
| ET-Flow (DRUGS RS) | 79.53 | 84.57 | 0.452 | 0.419 | 74.38 | 81.04 | 0.541 | 0.470 |
| ET-Flow (DRUGS SS) | 76.06 | 80.65 | 0.644 | 0.545 | 67.83 | 74.19 | 0.511 | 0.473 |

To further evaluate the generalization performance of ET-Flow, we conduct two more out-of-distribution experiments in addition to GEOM-XL. First, we test the model on scaffold-based splits of the GEOM-QM9 and GEOM-DRUGS dataset, which offers a more challenging alternative to the standard random split. We split the datasets based on Murcko scaffolds of the molecules into an 80:10:10 ratio for train, validation, and test sets. We evaluate our method on 1000 randomly sampled molecules from the resulting test set. The second experiment involves training the model on GEOM-DRUGS and assessing its performance on GEOM-QM9, a dataset with significantly smaller molecules. This experiment complements the generalization task to larger molecules in

GEOM-XL by assessing ability for ET-Flow to generalize to smaller molecules. The results, presented in Table 9, indicate that the model's performance degrades only marginally on the scaffold-based split. Furthermore, the model demonstrates robust performance on GEOM-QM9 even when trained on GEOM-DRUGS.

# E   Visualizations

Figure 6 shows randomly selected examples of sampled conformers from ET-Flow for GEOM-DRUGS. The left column is the reference molecule from the ground truth, and the remaining columns are samples generated with 50 sampling steps. Figure 7 showcases the ability for ET-Flow to generate quality samples with fewer sampling steps.

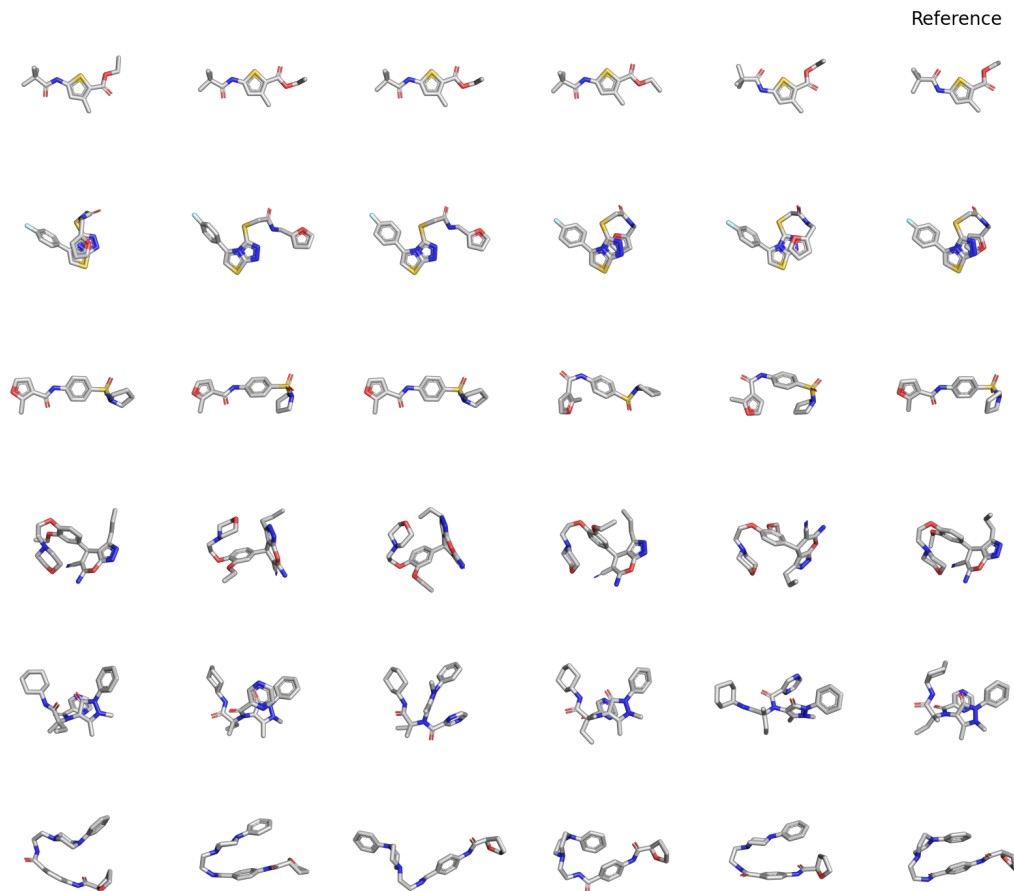

Figure 6: Examples of conformers of ground truth and ET-Flow for GEOM-DRUGS.

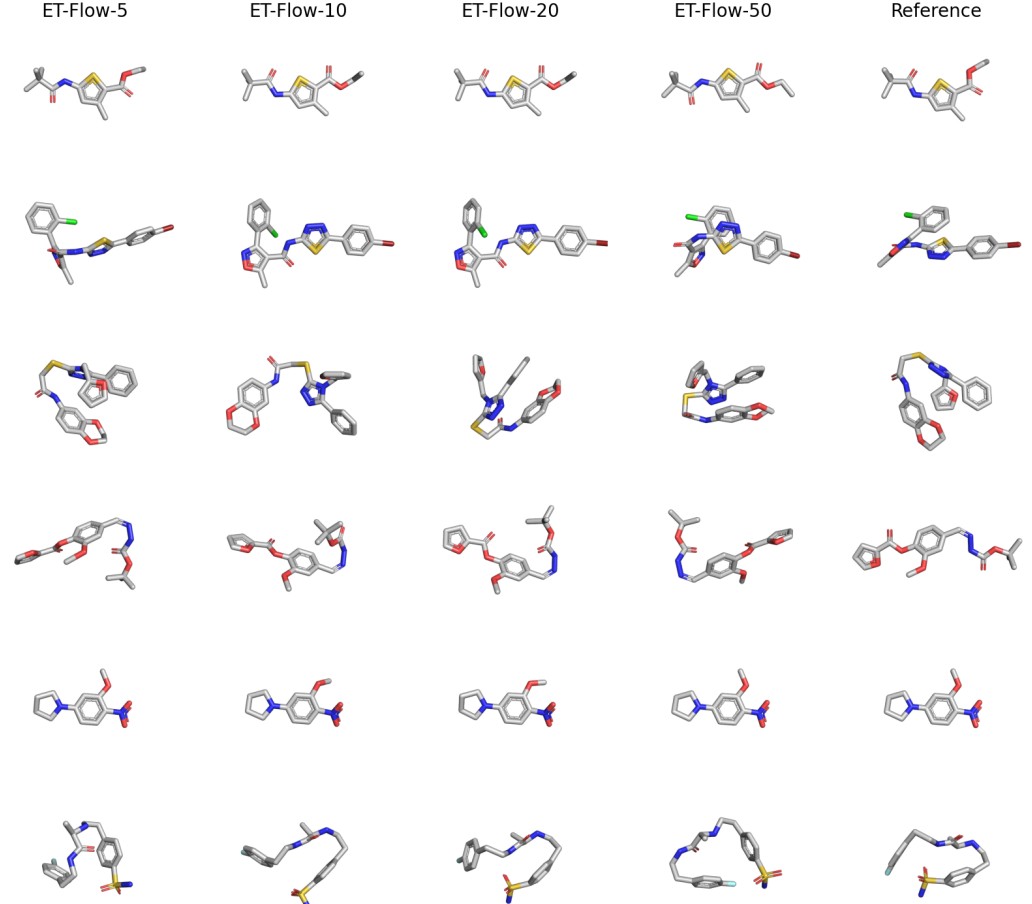

Figure 7: Examples of conformers of ground truth and ET-Flow for different number of sampling steps.

