# OpenReview forum: "ET-Flow: Equivariant Flow-Matching for Molecular Conformer Generation"
_NeurIPS.cc/2024/Conference — NeurIPS 2024 poster_

### Official Review · Reviewer_eBNM · 2024-07-10

**Soundness:** 3
**Presentation:** 3
**Contribution:** 2
**Rating:** 4
**Confidence:** 4

**Summary:**

The paper proposes the Equivariant Transformer Flow (ET-Flow) to generate high-quality molecule conformations. The authors use rotational alignment, stochastic sampling, and chirality correction to improve the flow matching framework for this task. Additionally, the paper modifies the TorchMD-NET equivariant transformer architecture to parameterize the target vector field. Experiments on the GEOM dataset show that  ET-Flow can perform well on the conformation prediction task. Furthermore, conformations generated by ET-Flow can also be used to predict ensemble properties with great performance.

**Strengths:**

1. Very clear writing.
2. The method is simple and easy to understand.

**Weaknesses:**

1. Lack of experiments. The experiments on the GEOM dataset are not enough to show the empirical performance of the framework. Some further experiments on large-scale datasets with more data and larger system size are necessary. I suggest the authors do experiments on OC20/OC22(Open Catalyst 2020/2022) datasets to test the performance of the ET-Flow framework.
2. The novelty of this work is not enough. I think the idea and method of this paper is very similar to [1], but in this work, the authors use a transformer-based model. Additionally, this paper proposes several tricks such as rotational alignment, stochastic sampling, and chirality correction. So the authors should clarify the novelty of this work.

[1]. Klein, Leon, Andreas Krämer, and Frank Noé. "Equivariant flow matching." Advances in Neural Information Processing Systems 36 (2024).

**Questions:**

See the sections above.

---

> ### Author Rebuttal · Authors · 2024-08-06
>
> > **Lack of experiments. The experiments on the GEOM dataset are not enough to show the empirical performance of the framework. Some further experiments on large-scale datasets with more data and larger system size are necessary. I suggest the authors do experiments on OC20/OC22(Open Catalyst 2020/2022) datasets to test the performance of the ET-Flow framework.**
>
> Thank you for your valuable feedback. Our primary focus in this work is on sampling molecular conformers, and therefore, the dataset and benchmark metrics we used align with the standards established by previous works that addressed the same objective from different perspectives[1,2,3]. While we appreciate the suggestion to test our framework on the OC20/OC22 datasets, this would be a nontrivial task due to the fundamentally different nature of the catalyst datasets. Consequently, existing methods for molecular conformer generation do not benchmark performance on OC20/22 and therefore is beyond the scope of our work. However, we agree that evaluating our framework on a diverse set of datasets is an important and promising direction for future research.
>
> > **The novelty of this work is not enough. I think the idea and method of this paper is very similar to [1], but in this work, the authors use a transformer-based model. Additionally, this paper proposes several tricks such as rotational alignment, stochastic sampling, and chirality correction. So the authors should clarify the novelty of this work.**
>
> We argue that the correct integration and modification of existing methodologies to achieve significant improvements should not be overlooked. One of the strengths of our work lies in identifying the core problem with existing methods and devising a straightforward approach with precise engineering, which is often crucial for empirically important tasks, especially in the application of machine learning to chemistry. We include ablation studies in table 2 of the global rebuttal demonstrating the effect of each design choice on the result.
>
> [1] Ganea, O., Pattanaik, L., Coley, C., Barzilay, R., Jensen, K., Green, W. and Jaakkola, T., 2021. Geomol: Torsional geometric generation of molecular 3d conformer ensembles. Advances in Neural Information Processing Systems, 34, pp.13757-13769.
>
> [2] Jing, B., Corso, G., Chang, J., Barzilay, R. and Jaakkola, T., 2022. Torsional diffusion for molecular conformer generation. Advances in Neural Information Processing Systems, 35, pp.24240-24253.
>
> [3] Wang, Y., Elhag, A. A., Jaitly, N., Susskind, J. M., & Bautista, M. Á. Swallowing the Bitter Pill: Simplified Scalable Conformer Generation. In Forty-first International Conference on Machine Learning.

---

> > ### Comment · Reviewer_eBNM · 2024-08-11
> >
> > Thanks for your reply.
> >
> > As you have mentioned in the paper, your goal is to construct a "scalable equivariant model that generates energy-minimized conformers given a molecular graph". I think you can test your framework in the OC20/22 datasets because the
> > IS2RS task (predict the relaxation structure based on the initial structure) is equivalent to "generate an energy-minimized conformer".  Indeed, it is a nontrivial task but it's necessary to test your method on large-scale datasets with more data and larger system size.
> >
> > I thank you again for clarifying the novelty of your work.

---

> > > ### Author Response · Authors · 2024-08-12
> > >
> > > Thanks for the nice suggestion! In OC20, one aims to find the relaxed structure of a molecule conditioned on a slab, which differs fundamentally from our work. The catalyst periodicity requires specialized approaches, such as AdsorbDiff[1].
> > >
> > > To compare the OC20 dataset size and diversity with our datasets:
> > > - The GEOM-DRUGS training dataset contains ~240,000 molecules and up to 30 conformations each, resulting in 5.7 million conformations. OC20 contains 82 molecules.
> > > - GEOM-DRUGS molecules (\~44 atoms) are larger than the average size of the adsorbates (structure to be relaxed) in OC20 (\~5 atoms)
> > > - To evaluate on larger molecules, we also evaluate on GEOM-XL (>100 atoms) with larger molecules than the training distribution. The results are shown in Appendix C.1 table 7. Do the experiments on Geom-XL address your concerns sufficiently under these considerations?
> > >
> > > We appreciate your insights and look forward to your feedback on this approach.
> > >
> > > [1] Kolluru, A., & Kitchin, J. R., 2024. AdsorbDiff: Adsorbate Placement via Conditional Denoising Diffusion. arXiv preprint arXiv:2405.03962.

---

### Official Review · Reviewer_AAmo · 2024-07-13

**Soundness:** 3
**Presentation:** 3
**Contribution:** 3
**Rating:** 7
**Confidence:** 4

**Summary:**

The paper proposes the Equivariant Transformer Flow (ET-Flow) which predicts low-energy molecular conformations given the molecular graphs. Unlike existing methods that rely on large transformer-based models for conformed fields or complex internal geometry calculations, ET-Flow leverages flow matching with equivariance and harmonic prior which directly operates on all-atom coordinates with minimal assumptions. The extensive experimental results illustrate that ET-Flow achieves state-of-the-art performance in molecular conformer generation benchmarks with fewer parameters and faster inference times, outperforming or matching previous methods while maintaining high accuracy and efficiency.

**Strengths:**

1. The proposed approach represents a significant innovation by leveraging flow matching with equivariance and harmonic prior in order to simplify the conformer generation process while maintaining high accuracy.

1. ET-Flow achieves SOTA model performance on molecular conformer generation benchmarks, outperforming or matching existing methods with much fewer parameters and faster inference times without sacrificing accuracy. The high accuracy and efficiency make it a promising candidate for practical applications.

1. The paper provides comprehensive experimental results, comparing ET-Flow with several leading approaches across various datasets. The inclusion of various evaluation metrics strengthens the validity of the findings.

**Weaknesses:**

1. ET-Flow achieves SOTA or near-SOTA performance on the benchmarks. However, my concern is that TorchMD-Net is already a strong model with a well-established architecture that leverages equivariant transformers for molecular modeling. Given this, it can be challenging to attribute the performance improvements of ET-Flow to the flow matching approach. Ablation studies focusing on the flow matching component or results from simpler architectures might be useful.

1. The need for a post hoc chirality correction step suggests that the model does not inherently handle stereochemistry (baseline models like MCF do not require such an explicit correction). Such a weakness may result in issues in practical applications.

1. ET-Flow primarily combines TorchMD-Net with the flow matching technique. While both components are robust and effective, the novelty of the approach is limited as it largely builds on existing methodologies. This integration, although leading to performance improvements, does not significantly advance the state-of-the-art in terms of methodological innovation.

**Questions:**

1. The proposed model utilizes an equivariant model architecture. Meanwhile, it is widely argued that equivariance is not a requirement for molecular machine learning (as suggested in the MCF paper). Without equivariance, the model is more easily generalized and scaled, and the symmetry could be implicitly learned from the data. I would like to ask the authors about the thoughts on this. In particular, I'm interested in the scaling performance (with respect to data or model size) of ET-Flow compared to MCF.

1. Have the authors considered conducting data augmentation to improve the diversity of generated conformations and enhance recall metrics?

**Limitations:**

The authors have discussed the limitations of recall performance and additional chirality correction steps. No potential negative societal impact is involved.

---

> ### Author Rebuttal · Authors · 2024-08-06
>
> Dear Reviewer,
>
> We first want to thank the reviewer for taking the time to review our work and ask thought-provoking questions. We hope that we are able to address said questions and concerns below.
>
> > **It can be challenging to attribute the performance improvements of ET-Flow to the flow matching approach. Ablation studies focusing on the flow matching component or results from simpler architectures might be useful.**
>
> We first want to acknowledge that the performance of our model is not only attributed to the use of flow matching but is a result of a combination of different components such as choice of prior, rotational alignment, sampling method. We additionally included ablation studies demonstrating the effect of these design choices in table 2 in global rebuttal.
>
> We agree that leveraging the expressive equivariant architecture combined with apt modification and engineering is indeed one of the key reasons for improved performance. However, the strengths of our approach, such as faster inference and flexible choice of prior are enabled by adopting the flow matching framework. Therefore, we view our work as addressing different challenges to create a harmonious integration of these components.
>
> > **The need for a post hoc chirality correction step. Such a weakness may result in issues in practical applications.**
>
> TorchMDNet is O(3) equivariant, thus necessitating a post-hoc chirality correction (CC) step to break the reflection symmetry. We conduct an additional experiment with SO(3) equivariance, alleviating the need for an additional correction step. This is achieved by modifying the vector output of TorchMDNet using the cross product in Equation(18). The empirical results are shown in table 3 for GEOM-DRUGs and table 4 for GEOM-QM9 in global rebuttal.
>
> For GEOM-QM9, we show that the O(3) with CC and SO(3) TorchMDNET achieve roughly similar results, however in the case of GEOM-DRUGS the advantage of O(3) with CC is more pronounced. An important point to highlight is the negligible cost of the CC step as highlighted in figure 1 in the global rebuttal pdf. In practice, one may choose to use ET-Flow combined with CC or directly use the SO(3) version of the model.
>
> While we acknowledge the reviewer's concern, we respectfully disagree about the practical weaknesses. MCF uses data augmentation and more model parameters to learn physical symmetries. As shown in figure 1 of the attached PDF in the global rebuttal, our method’s inference time remains significantly faster, even with the CC step. Furthermore, ET-Flow’s inference time primarily depends on the number of parameters.
>
> > **The novelty of the approach is limited as it largely builds on existing methodologies. This integration, although leading to performance improvements, does not significantly advance the state-of-the-art in terms of methodological innovation.**
>
> We argue that the correct integration and modification of existing methodologies to achieve significant improvements should not be overlooked. One of the strengths of our work lies in identifying the core problem with existing methods and devising a straightforward approach with precise engineering, which is often crucial for empirically important tasks, especially in the application of machine learning to chemistry. We also conduct design choice ablations as shown in Table 2 in the global rebuttal.
>
> > **It is widely argued that equivariance is not a requirement for molecular machine learning. Without equivariance, the model is more easily generalized and scaled, and the symmetry could be implicitly learned from the data. I'm interested in the scaling performance (with respect to data or model size) of ET-Flow compared to MCF.**
>
> The MCF paper advocates for a more general approach to molecular generative tasks by not embedding geometric inductive biases into the model, which is a valid and valuable perspective. However, our stance is that in the application of AI to scientific domains, leveraging known inductive biases is crucial for effective performance (scale vs performance tradeoff). Explicitly incorporating inductive biases ensures that the model inherently respects physical symmetries, leading to more reliable and physically consistent predictions. This approach is particularly important in domains where data can be scarce or expensive to obtain, as the inductive biases help the model generalize better from limited information.
>
> In our work, we used a dataset of comparable size to MCF but kept our model parameters(8.3M) significantly smaller. Although our computational limitations prevented us from scaling up to a similar number of parameters as used in MCF-L(242M), this is an avenue we plan to explore in the near future. Despite these limitations, our approach demonstrates the effective use of equivariant models in achieving competitive results, suggesting that incorporating inductive biases can lead to more efficient and scalable solutions in molecular machine learning.
>
> > **Have the authors considered conducting data augmentation to enhance diversity and recall metrics?**
>
> We believe this suggestion may be coming from the context of its use in MCF and AlphaFold3. In their cases, data augmentation was primarily employed to compensate for the lack of inductive bias, enabling the models to be more expressive while learning equivariance in a soft manner. In contrast, our model inherently incorporates symmetries by design, eliminating the need for such techniques. Additionally, while our recall metrics are slightly lower than those of MCF-M and MCF-L, our precision metrics show a significant improvement over all versions of MCF. In practice, this improvement in precision is crucial because this produces more physically-correct molecules as evidenced in our ensemble properties table having the lowest errors amongst the existing methods.

---

> > ### Comment · Reviewer_AAmo · 2024-08-12
> >
> > Thank the authors for the response. The additional experimental results and thorough analysis provided in the rebuttal effectively address most of my concerns and strongly support the paper's ideas. With that, I'd like to raise my score.

---

> > > ### Author Response · Authors · 2024-08-12
> > >
> > > We sincerely thank the reviewer for their thoughtful feedback and insightful discussion. This has greatly contributed to enhancing the quality of our work.

---

### Official Review · Reviewer_281V · 2024-07-16

**Soundness:** 3
**Presentation:** 4
**Contribution:** 2
**Rating:** 6
**Confidence:** 4

**Summary:**

The paper describes an equivariant flow matching model for conformer generation.  The stated contributions are accurate - the model performance is state-of-the-art and largely due to good engineering.

**Strengths:**

The paper is well written and the evaluations follow other published work.  Informative ablation studies are performed.  The evaluation of ensemble property averages is particularly appreciated.

I appreciate the authors highlight the modifications required for stable training.

**Weaknesses:**

Very little evaluation of out-of-distribution performance is considered (a small evaluation is in the appendix).  The ensemble property averages evaluation seems to be done on molecules drawn from the training data (if this is not the case, it needs to be more clear).  The paper would be stronger if generalization performance was more comprehensively evaluated.

Although efficiency is one of the main claims, there's no evaluation of inference time.

I think "DRUGS" is used as a short-hand for GEOM-DRUGS - this is confusing, use consistent nomenclature throughout.

**Questions:**

What is the out-of-distribution performance?  How well does the model generalize to different chemotypes (not just larger) and how does this compare to conventional and other generative models?

It would be interesting to see MCF trained using the same frame work as ETflow - e.g. how much of the performance is due to the choices in Table 6 versus the model architecture.

Can you perform evaluations that demonstrate these conformers are better for downstream tasks than those generated through conventional approaches?

**Limitations:**

Some recent work has pointed to the fact that improved recall/precision on these GEOM benchmarks does not necessarily result in better performance on downstream tasks that use conformers.  At least some discussion of this limitation would be appreciated.

---

> ### Author Rebuttal · Authors · 2024-08-06
>
> Dear Reviewer,
>
> We first want to thank the reviewer for taking the time to review our work and ask thought-provoking questions. We hope that we are able to address said questions and concerns below.
>
> > **Very little evaluation of out-of-distribution performance is considered (a small evaluation is in the appendix).**
>
> > **The paper would be stronger if generalization performance was more comprehensively evaluated.**
>
> > **What is the out-of-distribution performance?**
>
> We first want to acknowledge that we followed the evaluation protocols as done with prior work [1, 2, 3]. In the prior works, out-of-distribution (OOD) evaluation is done on GEOM-XL (Appendix C.1 table 7), which consists of molecules >100 atoms, using a model trained on GEOM-DRUGS (average 44 atoms per molecule). We demonstrate that our method performs slightly better than the MCF-S while having 5 million fewer parameters. However, we agree that more experiments that evaluate OOD performance is needed. Accordingly, we also evaluate our model on molecules significantly smaller than the training distribution via GEOM-QM9 (average 18 atoms per molecule) [4]. We provide these results in the table 1 of the global rebuttal statement. We are open to more suggestions and feedback regarding evaluation strategies.
>
> > **How well does the model generalize to different chemotypes (not just larger) and how does this compare to conventional and other generative models?**
>
> Thank you for bringing this to our attention. The idea of exploring different chemotypes is an interesting suggestion, but it is unclear what chemotypes are being referenced. More details about this would be appreciated so we could discuss further. Moreover, we would like to make clear that our domain of application is in drug discovery and therefore we focus on drug like molecules from the GEOM dataset [4].
>
> > **The ensemble property averages evaluation seems to be done on molecules drawn from the training data (if this is not the case, it needs to be more clear).**
>
> We would like to make clear that the ensemble averages experiment was done on the 100 randomly sampled molecules **from the test set**. We added a statement to section 4.4 to make that more clear.
>
> > **Although efficiency is one of the main claims, there's no evaluation of inference time.**
>
> Thank you for bringing this up. We included a plot of inference time with respect to the number of steps in figure 1 in the attached pdf for global rebuttal. We also include a plot on performance (precision) with respect to inference time in figure 2 in the attached pdf for global rebuttal.
>
> > **I think "DRUGS" is used as a short-hand for GEOM-DRUGS - this is confusing, use consistent nomenclature throughout.**
>
> We made the adjustment to the instances where we use “DRUGS” and just use “GEOM-DRUGS” for consistency.
>
> > **It would be interesting to see MCF trained using the same frame work as ETflow - e.g. how much of the performance is due to the choices in Table 6 versus the model architecture.**
>
> MCF follows a different framework than ETFlow. We would like to ask if the reviewer is implying to evaluate the architecture of MCF (Perceiver IO) with the flow matching framework? Any clarifications on this statement would be much appreciated, thank you.
>
> > **Some recent work has pointed to the fact that improved recall/precision on these GEOM benchmarks does not necessarily result in better performance on downstream tasks that use conformers. At least some discussion of this limitation would be appreciated.**
>
> We would like to point out to the reviewer that the ensemble properties table in Section 4.4 - table 3 in the manuscript indicates that our method has the lowest errors for the chemical properties evaluated against GFN-xTB oracle, demonstrating our model produces more physically-correct molecules compared to other methods. We would be open to benchmarking our model on other metrics if the reviewer could refer us to said metrics and alternative downstream evaluations.
>
> [1] Ganea, O., Pattanaik, L., Coley, C., Barzilay, R., Jensen, K., Green, W. and Jaakkola, T., 2021. Geomol: Torsional geometric generation of molecular 3d conformer ensembles. Advances in Neural Information Processing Systems, 34, pp.13757-13769.
>
> [2] Jing, B., Corso, G., Chang, J., Barzilay, R. and Jaakkola, T., 2022. Torsional diffusion for molecular conformer generation. Advances in Neural Information Processing Systems, 35, pp.24240-24253.
>
> [3] Wang, Y., Elhag, A. A., Jaitly, N., Susskind, J. M., & Bautista, M. Á. Swallowing the Bitter Pill: Simplified Scalable Conformer Generation. In Forty-first International Conference on Machine Learning.
>
> [4] Axelrod, S. and Gomez-Bombarelli, R., 2022. GEOM, energy-annotated molecular conformations for property prediction and molecular generation. Scientific Data, 9(1), p.185.

---

> > ### Comment · Reviewer_281V · 2024-08-07
> >
> > The additional evaluations (out of distribution, inference time, and ablation studies) definitely strengthen the paper (assuming they get included).
> >
> > A way to evaluate generalization to novel chemotypes is to perform a scaffold split of the training data.
> >
> > Downstream tasks from conformer generation include molecular docking, shape similarity, and pharmacophore search.  While I would not expect a comprehensive evaluation of ET-Flow's performance in these tasks in this paper, if the goal of the authors is to develop methods that advance drug discovery and not just to get a bold number in an ML conference paper, they should be evaluated and compared to standard approaches (e.g. RDKit ensembles - see https://pubs.acs.org/doi/full/10.1021/acs.jcim.3c01245).

---

> > > ### Author Response · Authors · 2024-08-12
> > >
> > > We would like to thank the reviewer for the valuable feedback. We will indeed incorporate the additional evaluations into the manuscript. Currently, we are conducting an experiment on GEOM-QM9 using a scaffold split, and we expect to upload the results within a day. We also plan to run a scaffold split experiment on GEOM-DRUGS, but given time and compute constraints, we will not be able to provide those results immediately. We will include both scaffold-split experiments in the manuscript.
> > >
> > > Are there any additional experiments or discussions the reviewer would recommend to further enhance the quality of the paper?

---

> ### Author Response · Authors · 2024-08-13
>
> For the scaffold split experiment, we divide the GEOM-QM9 smiles based on molecular scaffolds into an 80:10:10 ratio for train, validation, and test sets.  Our results based on 1000 randomly sampled molecules from the test set are as follows,
>
> | Method | Recall Coverage (mean) | Recall Coverage (median) | Recall AMR (mean) | Recall AMR (median) | Precision Coverage (mean) | Precision Coverage (median) | Precision AMR (mean) | Precision AMR (median) |
> |--------|------------------------|--------------------------|-------------------|---------------------|---------------------------|-----------------------------|-----------------------|------------------------|
> | ET-Flow (Random Split) | 94.99 | 100.00 | 0.083 | 0.035 | 91.00 | 100.00 | 0.116 | 0.047 |
> | ET-Flow (Scaffold Split) | 95.00 | 100.00 | 0.083 | 0.029 | 90.25 | 100.00 | 0.124 | 0.053 |
>
> It seems that generalization across scaffolds is possible. We will include the same experiments on GEOM-DRUGS, which are still running.
>
> Thanks for the suggestion - this experiment is a nice addition! Please let us know if they sufficiently address your concern.

---

### Author Rebuttal · Authors · 2024-08-06

We perform these additional experiments to support our rebuttals.

### **Out-of-Distribution Evaluation on GEOM-QM9**
To improve upon out-of-distribution evaluation, we test our model trained on GEOM-DRUGS on GEOM-QM9 (significantly smaller molecules).

| Method | Recall Coverage (mean) | Recall Coverage (median) | Recall AMR (mean) | Recall AMR (median) | Precision Coverage (mean) | Precision Coverage (median) | Precision AMR (mean) | Precision AMR (median) |
|--------|------------------------|--------------------------|-------------------|---------------------|---------------------------|-----------------------------|-----------------------|------------------------|
| CGCF | 69.47 | 96.15 | 0.425 | 0.374 | 38.20 | 33.33 | 0.711 | 0.695 |
| GeoDiff | 76.50 | **100.00** | 0.297 | 0.229 | 50.00 | 33.50 | 1.524 | 0.510 |
| GeoMol | 91.50 | **100.00** | 0.225 | 0.193 | 87.60 | **100.00** | 0.270 | 0.241 |
| Torsional Diff. | 92.80 | **100.00** | 0.178 | 0.147 | 92.70 | **100.00** | 0.221 | 0.195 |
| MCF | **95.0** | **100.00** | 0.103 | 0.044 | **93.7** | **100.00** | 0.119 | 0.055 |
| ET-Flow | **94.99** | **100.00** | **0.083** | **0.035** | 91.00 | **100.00** | **0.116** | **0.047** |
| ET-Flow-OOD | 86.68 | **100.00** | 0.218 | 0.160 | 68.69 | 75.3 | 0.369 | 0.317 |

Table 1: Molecule conformer generation results on GEOM-QM9 (δ = 0.5Å). For our method, we sample conformations over 50 time-steps. Bold indicates best performance. ET-Flow-OOD is the model trained on GEOM-DRUGS and tested on GEOM-QM9.

### **Design Choice Ablation**

We conduct a series of ablation studies to assess the influence of each component in the ET-Flow. Particularly, we re-run the experiments with (1) $O(3)$ equivariance without chirality correction, (2) Absence of Alignment, (3) Gaussian Prior as a base distribution. We demonstrate that improving probability paths and utilizing an expressive equivariant architecture with correct symmetries are key components for ET-Flow to achieve state of the art performance. The ablations were ran with reduced settings ($50$ epochs; lr $=1e-4$; $4$ A100 gpus).

| Method | Recall Coverage (mean) | Recall Coverage (median) | Recall AMR (mean) | Recall AMR (median) | Precision Coverage (mean) | Precision Coverage (median) | Precision AMR (mean) | Precision AMR (median) |
|--------|------------------------|--------------------------|-------------------|---------------------|---------------------------|-----------------------------|-----------------------|------------------------|
| Our Method | 75.37 | 82.35 | 0.557 | 0.529 | 58.90 | 60.87 | 0.742 | 0.690 |
| Our Method (O(3)) | 72.74 | 79.21 | 0.576 | 0.556 | 54.84 | 54.11 | 0.794 | 0.739 |
| Our Method (w/o Alignment) | 68.67 | 74.71 | 0.622 | 0.611 | 47.09 | 44.25 | 0.870 | 0.832 |
| Our Method (Gaussian Prior) | 66.53 | 73.01 | 0.640 | 0.625 | 44.41 | 40.88 | 0.903 | 0.864 |

Table 2: Comparison of different variants of our method. Coverage (↑) is better when higher, AMR (↓) is better when lower.

### **Chirality correction and SO(3) Study**

We conduct ablation experiments evaluating ET-Flow with and without Chirality Correction (CC). Additionally, we also report performance with an SO(3) equivariant version of ET-Flow without Chirality correction.

| Method | Recall Coverage (mean) | Recall Coverage (median) | Recall AMR (mean) | Recall AMR (median) | Precision Coverage (mean) | Precision Coverage (median) | Precision AMR (mean) | Precision AMR (median) |
|--------|------------------------|--------------------------|-------------------|---------------------|---------------------------|-----------------------------|-----------------------|------------------------|
| GeoDiff | 42.10 | 37.80 | 0.835 | 0.809 | 24.90 | 14.50 | 1.136 | 1.090 |
| GeoMol | 44.60 | 41.40 | 0.875 | 0.834 | 43.00 | 36.40 | 0.928 | 0.841 |
| Torsional Diff. | 72.70 | 80.00 | 0.582 | 0.565 | 55.20 | 56.90 | 0.778 | 0.729 |
| MCF - S (13M) | 79.4 | 87.5 | 0.512 | 0.492 | 57.4 | 57.6 | 0.761 | 0.715 |
| MCF - B (62M) | 84.0 | 91.5 | 0.427 | 0.402 | 64.0 | 66.2 | 0.667 | 0.605 |
| MCF - L (242M) | **84.7** | **92.2** | **0.390** | **0.247** | 66.8 | 71.3 | 0.618 | 0.530 |
| ET-Flow (8.3M) O(3) | 78.6 | 83.33 | 0.479 | 0.455 | 67.16 | 72.15 | 0.637 | 0.563 |
| ET-Flow (8.3M) O(3) + CC | 79.53 | 84.57 | 0.452 | 0.419 | 74.38 | 81.04 | 0.541 | 0.470 |
| ET-Flow (9.1M) SO(3)    | 78.18 | 83.33 | 0.48 | 0.459 | 67.27 | 71.15 | 0.637 | 0.567 |

Table 3: Molecule conformer generation results on GEOM-DRUGS ($\delta$ = 0.75Å). For all ET-Flow methods, we sample conformations over 50 time-steps. Bold indicates best performance.

| Method | Recall Coverage (mean) | Recall Coverage (median) | Recall AMR (mean) | Recall AMR (median) | Precision Coverage (mean) | Precision Coverage (median) | Precision AMR (mean) | Precision AMR (median) |
|--------|------------------------|--------------------------|-------------------|---------------------|---------------------------|-----------------------------|-----------------------|------------------------|
| CGCF | 69.47 | 96.15 | 0.425 | 0.374 | 38.20 | 33.33 | 0.711 | 0.695 |
| GeoDiff | 76.50 | **100.00** | 0.297 | 0.229 | 50.00 | 33.50 | 1.524 | 0.510 |
| GeoMol | 91.50 | **100.00** | 0.225 | 0.193 | 87.60 | **100.00** | 0.270 | 0.241 |
| Torsional Diff. | 92.80 | **100.00** | 0.178 | 0.147 | 92.70 | **100.00** | 0.221 | 0.195 |
| MCF | 95.0 | **100.00** | 0.103 | 0.044 | 93.7 | **100.00** | 0.119 | 0.055 |
| ET-Flow (8.3M) O(3) + CC | 94.99 | **100.00** | 0.083 | 0.035 | 91.00 | **100.00** | 0.116 | 0.047 |
| ET-Flow (9.1M) SO(3) | **95.98** | **100.00** | **0.076** | **0.030** | **94.05** | **100.00** | **0.098** | **0.039** |

Table 4: Molecule conformer generation results on GEOM-QM9 (δ = 0.5Å). For all ET-Flow methods, we sample conformations over 50 time-steps. Bold indicates best performance.

---

### Decision · Program_Chairs · 2024-09-25

**Decision:**

Accept (poster)

**Comment:**

The paper presents a conformer generation approach using flow matching. In this task, given a graph a generative model generates 3D configurations of molecules which  coincide with energy minima (conformers). The approach is of limited technical novelty, but a nice display of how a simple, well-engineered model can achieve state of the art results. During discussion the authors engage well with the concerns raise by the reviewers, in particular, they add a number of important ablations.